# Picosecond Laser Ablation of Polyhydroxyalkanoates (PHAs): Comparative Study of Neat and Blended Material Response

**DOI:** 10.3390/polym12010127

**Published:** 2020-01-05

**Authors:** Rocío Ortiz, Pooja Basnett, Ipsita Roy, Iban Quintana

**Affiliations:** 1Physics of Surfaces and Materials Unit, TEKNIKER, Iñaki Goenaga 5, 20600 Eibar, Spain; iban.quintana@tekniker.es; 2Applied Biotechnology Research Group, Department of Life Sciences, Faculty of Science and Technology, University of Westminster, London W1W 6UW, UK; p.basnett@westminster.ac.uk; 3Department of Materials Science and Engineering, University of Sheffield, North Campus, Broad Lane, Sheffield S3 7HQ, UK

**Keywords:** polyhydroxyalkanoates (PHAs), picosecond pulsed laser ablation, surface micro structuring

## Abstract

Polyhydroxyalkanoates (PHAs) have emerged as a promising biodegradable and biocompatible material for scaffold manufacturing in the tissue engineering field and food packaging. Surface modification is usually required to improve cell biocompatibility and/or reduce bacteria proliferation. Picosecond laser ablation was applied for surface micro structuring of short- and medium-chain length-PHAs and its blend. The response of each material as a function of laser energy and wavelength was analyzed. Picosecond pulsed laser modified the surface topography without affecting the material properties. UV wavelength irradiation showed halved ablation thresholds compared to visible (VIS) wavelength, revealing a greater photochemical nature of the ablation process at ultraviolet (UV) wavelength. Nevertheless, the ablation rate and, therefore, ablation efficiency did not show a clear dependence on beam wavelength. The different mechanical behavior of the considered PHAs did not lead to different ablation thresholds on each polymer at a constant wavelength, suggesting the interplay of the material mechanical parameters to equalize ablation thresholds. Blended-PHA showed a significant reduction in the ablation threshold under VIS irradiation respect to the neat PHAs. Picosecond ablation was proved to be a convenient technique for micro structuring of PHAs to generate surface microfeatures appropriate to influence cell behavior and improve the biocompatibility of scaffolds in tissue engineering.

## 1. Introduction

Polyhydroxyalkanoates (PHAs) are natural polymers produced by microorganisms, which can be easily degraded and absorbed by them in natural environments, without toxic effects in living cells or tissues [1,2,3,4,5,6]. These highly biocompatible polymers can be synthesized with different thermal, mechanical, and degradation properties depending on their monomer chain length, selection of microbial production strain, substrates, and processing [7,8,9]. In addition, they can be commercially produced in large amounts by fermentation processes of both gram-negative and -positive bacteria in carbon-rich environments. More important, PHAs are thermoplastic polyesters liable to be processed by thermoforming techniques as hot embossing or injection molding [9]. These characteristics made of these polymers an ideal material for scaffold manufacturing in tissue engineering or for food packaging, where new and improved biodegradable materials are needed to replace synthetic polymers obtained from petroleum resources.

Typically, surface modification of films or components produced as scaffolds or food packaging materials is required to improve their surface properties in terms of avoiding bacterial proliferation [10] and, in the case of tissue repair devices, promoting cell biocompatibility and controlling cell behavior [11]. In this regard, surface features in the µm or nm range can be applied in order to interact with cells and influence their behavior. In many cases, this topographical modification is intended to provide an additional surface characteristic respecting the material bulk properties and even the chemical and microstructural surface properties [12]. Laser micro-processing offers many advantages compared to other existing surface micro structuring technologies, such as versatility in terms of materials to be processed (almost any material) and geometries to be generated, the fact of being a single-step and contactless method, and the easy adaptation of the process for micropatterning of tubular or more complex sample shapes [13]. Laser ablation by CO2 and short-pulsed (nanoseconds) lasers has been applied before on polyhydroxy butyrate (P3HB) to cut the material or modify its surface or bulk properties [14,15,16,17,18]. However, ultrashort pulsed lasers (in the picosecond and the femtosecond range) allow to generate surface microfeatures with higher precision and minimal thermal and chemical impact on biodegradable and biocompatible polymers [19,20]. Here, the response of novel blends of short-chain length (scl-) and medium-chain length (mcl-) PHAs to ps pulsed laser ablation at wavelengths of 355 and 532 nm is reported for the first time. Single and multiple pulses will be applied on three types of PHAs with different chain length and composition (a short-chain length PHA, a medium chain length PHA, and the blend). The effect of laser wavelength and pulse energy in surface morphology and ablation threshold will be analyzed. This analysis will allow to study the surface integrity achieved, and the absorption mechanisms that occurred in each of the materials considered, as well as their correlation with the material properties. Additionally, the ablation rate for micromachined grooves will be analyzed in terms of pulse energy.

## 2. Materials and Methods

### 2.1. Materials

PHAs were produced by the University of Westminster (London, UK). Neat PHAs with short (C3-C5 monomer units, scl-PHA (P(3HB))) and medium (C6-C16 monomer units, mcl-PHA (poly(3-hydroxy octanoate-co-3-hydroxy decanoate) (P(3HO-3HD)) chain lengths were produced at large scale by means of a bioreactor with a capacity of 72 L using liquid bacterial culture. A standard purification protocol was applied to obtain ultra-pure PHAs. These materials were used to prepare films of neat scl-PHAs, mcl-PHAs, and the blend (P(3HB)/P(3HO-3HD)) (80scl-PHAs/20mcl-PHAs), by casting solution in Chloroform at a concentration of 5 wt % (Figure 1). These films were chemically characterized by gas chromatography-mass spectrometry (GC-MS) and thermally characterized by differential scanning calorimetry (DSC). The mechanical properties of the films were characterized by tensile tests.

### 2.2. Laser Irradiation

Neat and blended PHAs films were ablated by means of a picosecond pulse Nd:YVO_4_ laser (RAPID: Coherent, Münster, Germany) integrated in a micromachining workstation by 3D-Micromac. A detailed description of the experimental set-up can be found in a previous publication of our group [19]. Polymer films were irradiated with short-wavelength (355 and 532 nm) pulses. Craters were produced on the surface of the three considered PHAs by applying single pulses at a frequency of 10 kHz, which allows to achieve the highest pulse energy and scanning speed of 1000 mm/s. On the blend, pulse overlapping was also applied to generate grooves. The percentage of overlapping between pulses (*U*_d_) can be calculated by Equation (1) as a function of the frequency of emission of the pulses (*f*), the speed of the scanner to move over the substrate (*v*) and the laser spot diameter (*D*) (Figure 2).
*U*_d_ = (1 − *v*/(*f*∙*D*)) × 100%(1)

The material response to the laser pulses was assessed using scanning electron microscopy SEM (Karl Zeiss XB1540, Jena, Germany) and Fourier transform infrared (FTIR) spectroscopy (JASCO FT/IR 4700 LE) with an information depth of one micrometre. Crater depths and diameters were measured by mechanical stylus profilometry (Dektak 8, Veeco, Plainview, NY, USA) and confocal microscopy.

## 3. Results

### 3.1. Neat PHAs

Single-pulse craters with different pulse energy were produced on scl- (Figure 3) and mcl-PHAs (Figure 4) to determine the energy ablation threshold at wavelengths of 355 (UV) and 532 nm (VIS). On scl-PHA (Figure 3), crater formation was observed when pulse energy was close to 22 µJ, at UV wavelength (Figure 3b), and 17.5 µJ at VIS wavelength (Figure 3e). Craters were approximately round with no signs of surface swelling prior to material ablation (no rim of recast material around the ablation zone). On mcl-PHA (Figure 4), UV wavelength irradiation produced small holes when low energies were applied (Figure 4a), while round-like crater formation was observed above 60 µJ (Figure 4c). On the contrary, laser ablation at VIS wavelength generated irregular-shaped notches with increased size as increasing pulse energy and proper crater-like shape was not observed until pulse energy reached a value of 32.7 µJ.

The diameter (*D*) and depth (*d*) of craters produced on scl- and mcl-PHAs at both wavelengths were measured as a function of pulse energy (*E*) by SEM and profilometric evaluation. The spot size *ω*_0_ (beam radius measured at 1/e^2^ or beam waist) and the threshold energy *E*_th_ for crater formation (onset of ablation) were evaluated by fitting the crater diameter and the pulse energy to the following well-known equation [21]:(2)$$D_{abl}^{2}=2\omega_{0}^{2}\ln\left(\frac{E}{E_{\mathrm{th}}}\right)$$

Figure 5 shows the *D*^2^ measurements obtained by SEM on scl-PHAs as a function of the applied pulse energy. Regarding laser ablation at UV wavelength, the values of *ω*_0_ and *E*_th_ obtained from the fit function were (16.43 ± 0.09) µm and (4.0 ± 0.8) µJ, respectively. This energy threshold was much lower than that required for crater formation, as observed in Figure 1 (22 µJ), but close to the minimum energy value at which surface modification was first observed in the form of small notches (5 µJ). The same was observed in the case of laser ablation at VIS wavelength, where *E*_th_ (6.8 ± 0.8 µJ) was also lower than the energy at which proper crater formation was produced (17.5 µJ, Figure 4) and matched the energy at which small holes with diameters minor than 10 micrometres were generated. Regarding mcl-PHA (Figure 6), the calculated laser beam parameters were *ω*_0_ (22.6 ± 0.1) µm and *E*_th_ (9.4 ± 0.6) µJ for the UV wavelength and (19.62 ± 0.09) µm and (12.6 ± 1.2) µJ for the VIS wavelength. As observed in the scl-PHA, the energy thresholds obtained from the fit function were very similar to the minimum energies at which first signs of surface ablation were observed on the material surface (Figure 4a,d).

Crater depth (*d*) was characterized for all the range of pulse energies by profilometry and confocal microscopy. In order to study the nature of the absorption mechanisms occurred in the PHAs under laser irradiation at considered wavelengths, an effective absorption coefficient (*α*_eff_) can be estimated from the following logarithmic expression when single-photon absorption processes occur [22]:(3)$$d=\frac{1}{\alpha_{\mathrm{eff}}}\ln\left(\frac{E}{E_{\mathrm{th}}}\right)$$

In the case of scl-PHA, the crater depth was approximately constant at a value of 1 or 2 micrometres, for the UV wavelength (Figure 7a), and at a value of 2–3 micrometres for the VIS wavelength (Figure 7b). As shown in the SEM images in Figure 3, the craters produced on the material by the laser irradiation at both wavelengths showed a very porous and irregular surface. Although this made measuring crater depth quite challenging, no significant differences in crater depth were observed on this material, and data cannot be fit to Equation (3) in the energy range considered. When laser ablation at UV wavelength was applied on mcl-PHAs (Figure 8a), a slight increase of crater depth with pulse energy was observed, and data can be successfully fitted to Equation (3). The energy threshold obtained from this equation was 4 µJ (*E*_th_ = (4 ± 2) µJ, *α*_eff_ = (0.28 ± 0.07) µm^−1^), slightly below the value obtained from fitting the measured squared crater diameters to the Equation (2) (*E*_th_ = 9.4 µJ). When VIS wavelength was applied (Figure 8b), crater depth did not show again a clear dependence on pulse energy and *E*_th_ and *α*_eff_ could not be estimated. Crater depth showed high dispersion in the energy range considered, with values varying between 6 and 10 micrometres.

### 3.2. PHA Blend

After characterizing the laser ablation parameters on the scl- and mcl-PHAs, we analyzed the laser ablation absorption process on the blend formed by these two neat polymers (80 scl-PHA/20 mcl-PHA). Figure 9 shows SEM images of single pulse craters produced by varying the pulse energy at UV and VIS wavelengths. The energy ablation thresholds observed on the blend for both wavelengths (3–4 µJ) were below the values observed on the neat polymers. At energies of 3 µJ, we did not observe any change in surface topography on scl-PHA and mcl-PHA, while this value lead to small crater formation in the blend material. The blend material was highly porous, and, in some cases, it was difficult to distinguish between the pores on the material surface and the craters or holes produced by the laser irradiation (Figure 9d). Crater diameter as a function of laser energy was fitted to Equation (2) (Figure 10) obtaining a spot size and threshold energy of (16.58 ± 0.08) µm and (3.1 ± 0.4) µJ for the UV wavelength, and (13.96 ± 0.09) µm and (3.7 ± 0.6) µJ for the VIS wavelength. These values were equal to the energy values at which crater formation was first observed, as well as those provided by the Equation (3) (*E*_th_ = (2.9 ± 0.5) µJ, *α*_eff_ = (0.26 ± 0.02) µm^−1^ (UV), *E*_th_ = (3.3 ±1.1) µJ, *α*_eff_ = (0.23 ± 0.03) µm^−1^ (VIS)) (Figure 11). As in the case of neat polymers, squared crater diameters showed deviations from the fitting at high pulse energies.

### 3.3. Effects of Pulse Overlapping on the Blend

On the blend, the effect of pulse energy and overlapping was analysed in terms of groove dimensions and surface integrity by SEM analysis. Figure 12 shows grooves produced on the blend by overlapping laser pulses (90.9% overlapping) at a wavelength of 355 nm and a frequency of 100 kHz and different pulse energy. For pulse energies below 3 µJ, no surface modification was observed. At higher pulse energies, the depth and width of the grooves became larger, reaching a plateau of 33 µm in width and 17 µm in depth beyond energies of 6 µJ (Figure 12c). Some pores were observed within the laser-created channel (Figure 12a,b). The material itself was quite porous, and the laser seemed to reduce the pore density, likely due to the formation of recast material produced as a consequence of the high energy deposited on the material by pulse overlapping. This recast material increased from approximately 1 µm, at energies below 5.5 µJ, to 3 µm at energies beyond this value. The effect of laser irradiation on material surface was analysed by FTIR on 40 × 40 mm samples, considering one pristine (no laser-ablated) sample, and a laser-ablated sample applying the highest pulse energy to generate grooves of 33 µm in width, 17 µm in depth and 10 µm of spacing between grooves (Figure 13). Both grooved and pristine samples showed spectra with almost the same peaks but lower signal in the case of the laser-ablated film, likely due to the higher roughness of the grooved compared to the pristine sample. Just a small peak at 1722 cm^−1^ (highlighted in the FTIR spectrum by a rectangle), identified as the crystalline-phase band of the carbonyl group of the molecule [23], seemed to almost disappear and slightly move to the amorphous-phase band (at 1738 cm^−1^) in the grooved spectrum respect to the pristine spectrum, suggesting the occurrence of a slight amorphization of the material in the laser-ablated area.

## 4. Discussion

Picosecond-pulsed laser irradiation at UV and VIS wavelengths was successfully applied to ablate three different types of PHAs, proving the capability of ultra-short pulsed lasers to process these polymers at longer wavelengths than those required by short-pulsed lasers, which are in the middle- and far-UV regions [15,16,18]. The fluence applied here (Table 1) was much higher than that applied by Michaljanicova et al. [18] on P3HB to increase surface roughness and cause chemical and phase bulk changes via nanosecond excimer lasers at middle- and far-UV regions (F > 15 mJcm^−2^). However, no significant structural changes nor surface affectation was observed here when applying picosecond lasers at high laser fluencies (F > 1 Jcm^−2^). For shorter pulse length, the applied energy is limited in a smaller material volume, increasing the absorbed energy, and ablation can occur by direct vaporization without significant heating effects (photochemical ablation). This allowed to ablate PHAs at higher wavelengths and fluencies causing mainly surface topographical changes without affecting the bulk properties.

Ablation of polymers by ultra-short pulsed laser involves the interplay of several mechanisms of different nature, such as photochemical (bond breaking by laser photons), photothermal (polymer bond breaking by electronic excitation and thermalization) and photomechanical processes (mechanical fractures caused by the high thermoelastic pressure wave induced by laser irradiation) [24]. The extent of each type of mechanism depends on the laser and material properties. On both neat and blended-PHAs, the ablation threshold for the UV wavelength was about half of the calculated for the VIS wavelength (Table 1). Since, according to molecular simulation (MDS) studies, lower ablation thresholds are characteristic of photochemical ablation [25], ablation by UV wavelength could involve more photochemical mechanisms than ablation by VIS wavelength. This could be caused likely by the higher energy of UV photons compared to VIS photons. The effective absorption coefficient obtained by fitting the measured depth of the craters obtained on mcl-PHA (at UV wavelength) and the blend (at both VIS and UV wavelengths) to Equation (3) is about (0.26 ± 0.08) µm^−1^. This value is lower than 1 µm^−1^, indicating no significant occurrence of single-photon absorption processes [22] and, therefore, the involvement of multiphoton effects in laser ablation of these PHAs at both wavelengths. Typically, on polyesters, photon absorption leads to electronic excitation of the C=O band in the ester functional group of the polymer, which is followed by intra- and intermolecular conversion to heat or bond scission. The occurrence of multiphoton absorption is also supported by the fact that single-photon energy is insufficient for bond breaking of the chromophore in the ester group (C=O, *E* = 8 eV), both in the case of UV and VIS laser irradiation. The etched depth on mcl- and blended-PHAs at both UV and VIS wavelengths were similar and around 8–10 micrometers for the same range of pulse energies, in agreement with the obtained similar efficient absorption coefficients, and showing no wavelength dependence. The etched depth on scl-PHA was significantly lower (2–3 micrometers) than on the mcl- and blended-PHAs, unfortunately, since the efficient absorption coefficients on this material could not be calculated, no relationship can be established between this parameter and the etch rate. Our findings regarding the wavelength dependence of ablation threshold and rate contradict those reported in previous studies of picosecond pulse laser ablation on other polymers [25,26,27,28], where ablation rate increased for shorter wavelengths while ablation threshold did not show any clear dependence. Here, the ablation rate did not show a clear dependence on beam wavelength, and the ablation threshold was found to be lower for the shortest wavelength irradiation.

The role that photomechanical mechanisms play in ultrashort pulsed laser ablation of polymers, gives to the material mechanical properties a significant influence on the ablation parameters. In the stress confinement irradiation regime, the ablation threshold occurs when the stress exceeds the dynamic tensile strength of the material [29]. Scl- and mcl-PHAs, despite the fact of showing quite different crystallinity, ultimate tensile strength, and elongation at break (Table 2 and Table 3), showed similar ablation thresholds at the same wavelength (Table 1). While the higher crystallinity and low elongation at break of the scl-PHA (*ɛ*_b_ = 7%) should lead to a reduction of the polymer resistance under mechanical stress and, therefore, a lower ablation threshold compared to the mcl-PHA (*ɛ*_b_ = 920%), its higher ultimate tensile strength (*σ*_U_ = 36 MPa) acts in opposition. Thus, although more amorphous polymers, such as the mcl-PHA, should better dissipate the absorbed energy and accommodate the stress because of their heterogeneous microstructure and viscous flow behavior, here, the interplay of all the mentioned mechanical parameters seems to lead to the equalization of the ablation thresholds on both polymers.

Regarding the blended-PHA, it was observed that, while the ablation threshold under UV irradiation was similar for the three polymers, the one obtained under VIS irradiation was almost halved respect to both scl- and mcl-PHAs. The heterogenous structure of the blend compared to the homogeneous neat PHAs could lead to lower mechanical resistance of the blend under laser irradiation. This structural difference could be significant only for the VIS wavelength irradiation due to the more photothermal and photomechanical nature of the VIS compared to UV laser ablation.

Picosecond laser ablation allows to directly and precisely generate craters (with no signs of swelling prior to ablation) and grooves (with minimal redeposition of ablated material) on the three different types of natural polymers considered in this study. In addition, these topographical features showed variable dimensions in the range of typical cell size (1–100 µm), appropriate for their application in surface micro structuring of biocompatible and biodegradable scaffolds made of PHAs for tissue engineering applications. Furthermore, according to Ellis et al. [15], the slight polymer amorphization observed in the FTIR spectra of the grooved blended-PHA sample could even promote cell adhesion and proliferation, improving the material biocompatibility, although further experiments must be performed to obtain conclusive results at this respect.

## 5. Conclusions

The growing need of advanced biomaterials for scaffold manufacturing in the tissue engineering field or for food packaging makes necessary the development of new biomaterials and surface nano- and micro-technologies able to process those. Picosecond pulsed laser ablation has been applied here for the first time for surface micro structuring of PHAs. Picosecond pulses ablate and modify the topography of three types of PHAs with different thermal and mechanical properties with non-significant effects on the chemical and microstructural properties of these materials. These findings suggest photochemical ablation as the dominant mechanism during picosecond laser ablation of PHAs, especially when applying irradiation at 355 nm wavelength. In addition, picosecond pulsed laser demonstrated a wide PHA processing window than short pulse lasers. Microfeatures in the form of craters and grooves with tunable dimensions within the range of typical cell sizes have been directly and precisely generated on PHAs in one-step process with minimal thermal impact on the material surface.

## Figures and Tables

**Figure 1 polymers-12-00127-f001:**
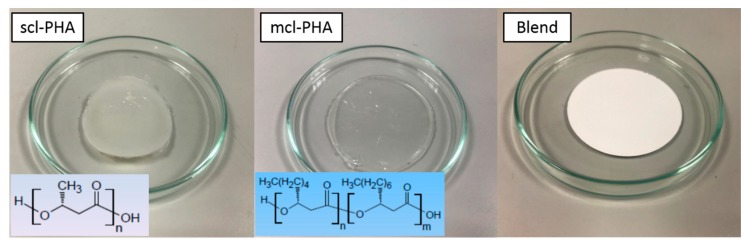
PHAs films prepared by casting solvent of scl- (P3HB) and mcl-PHAs (P(3HO-3HD) and their blend (P(3HB)/P(3HO-3HD) 80/20).

**Figure 2 polymers-12-00127-f002:**
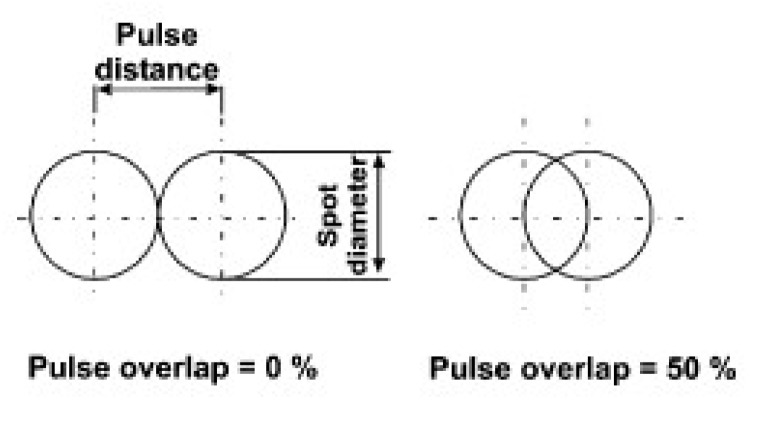
Drawing representing the pulse overlapping.

**Figure 3 polymers-12-00127-f003:**
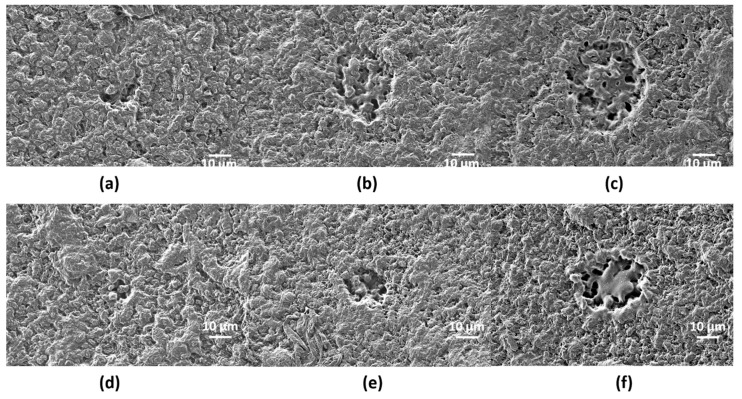
SEM images corresponding to single-shot craters at wavelengths of 355 nm (**a**–**c**) and 532 nm (**d**–**f**) on scl-PHA at different pulse energies: (**a**) *E* = 5 µJ, (**b**) *E* = 22 µJ, (**c**) *E* = 39.5 µJ, (**d**) *E* = 6.3 µJ, (**e**) *E* = 17.5 µJ, (**f**) *E* = 83.5 µJ.

**Figure 4 polymers-12-00127-f004:**
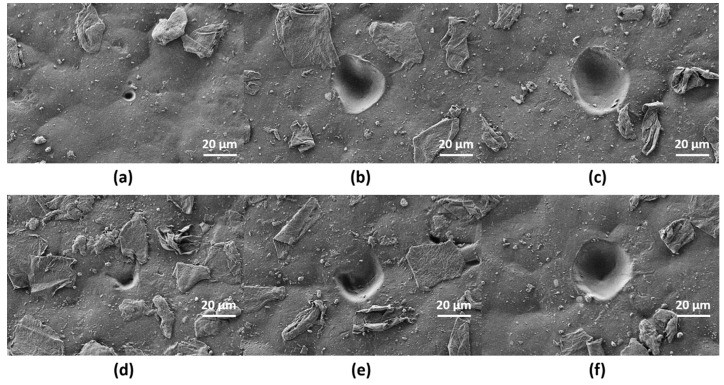
SEM images corresponding to single-shot craters at wavelengths of 355 nm (**a**–**c**) and 532 nm (**d**–**f**) on mcl-PHA at different energies: (**a**) *E* = 10 µJ, (**b**) *E* = 22 µJ, (**c**) *E* = 32.7 µJ, (**d**) *E* = 13 µJ, (**e**) *E* = 50 µJ, (**f**) *E* = 61 µJ.

**Figure 5 polymers-12-00127-f005:**
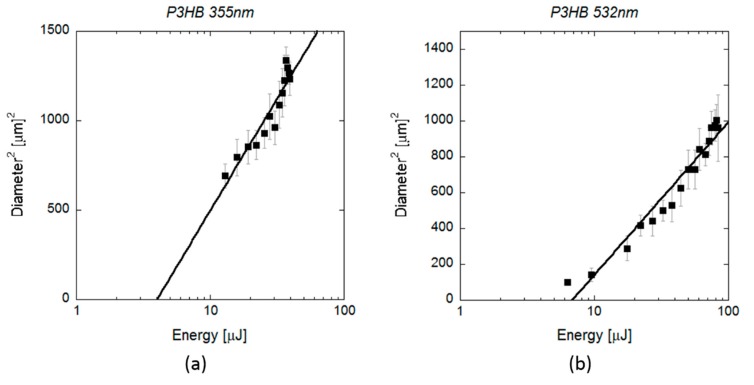
(**a**) Squared diameter of the craters (black rectangles) produced by 10 ps pulses at a wavelength of 355 and (**b**) 532 nm on scl-PHA and measured by SEM as a function of the pulse energy. Lines indicate the fitting curves to Equation (2): (**a**) *ω*_0_ = (16.43 ± 0.09) µm and *E*_th_ = (4.0 ± 0.8) µJ, (**b**) *ω*_0_ = (13.60 ± 0.16) µm and *E*_th_ = (6.8 ± 0.8) µJ.

**Figure 6 polymers-12-00127-f006:**
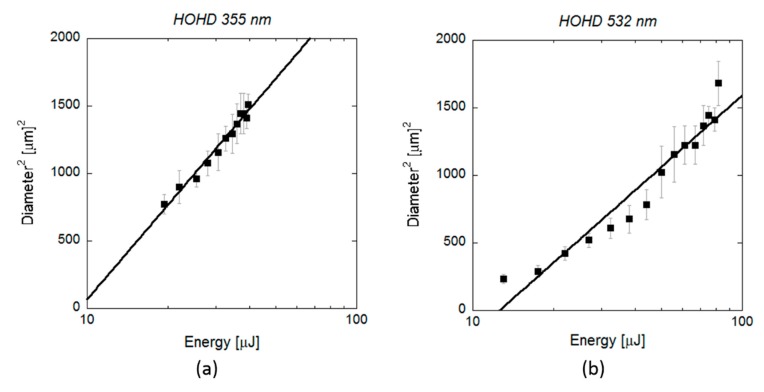
(**a**) Squared diameter of the craters (black rectangles) produced by 10 ps pulses at a wavelength of 355 and (**b**) 532 nm on mcl-PHA and measured by SEM as function of the pulse energy. Lines indicate the fitting curves to Equation (2): (**a**) *ω*_0_ = (22.6 ± 0.1) µm and *E*_th_ = (9.4 ± 0.6) µJ, (**b**) *ω*_0_ = (19.62 ± 0.09) µm and *E*_th_ = (12.6 ± 1.2) µJ.

**Figure 7 polymers-12-00127-f007:**
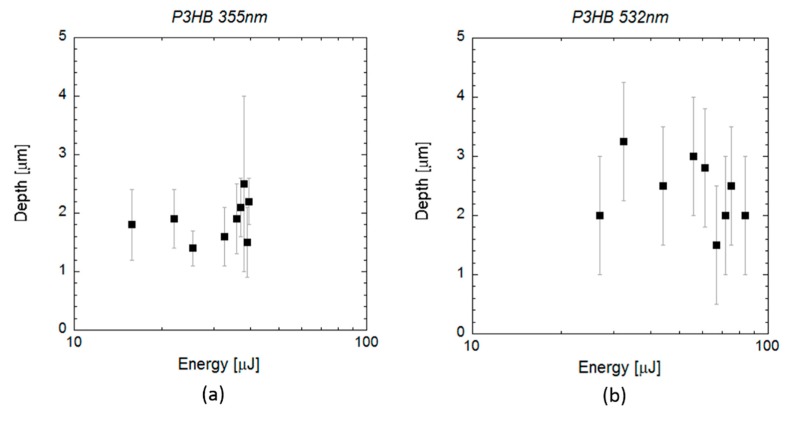
Data analysis of single pulse craters produced by 10 ps pulses at a wavelength of 355 and 532 nm on scl-PHA. The graphs show the depth of the craters (black rectangles) measured by profilometry and confocal microscopy (**a**) for wavelengths of 355 nm and (**b**) 532 nm as a function of the pulse energy.

**Figure 8 polymers-12-00127-f008:**
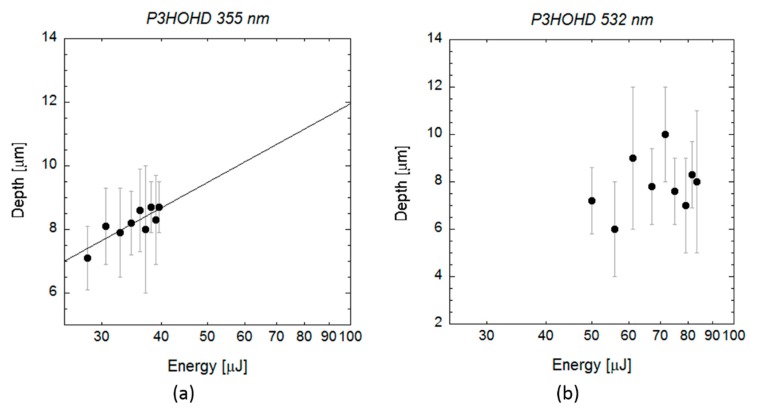
Data analysis of single pulse craters produced by 10 ps pulses (**a**) at a wavelength of 355 and (**b**) 532 nm on mcl-PHA. The graphs show the depth of the craters measured by profilometry (black circles) (**a**) for wavelengths of 355 nm and (**b**) 532 nm as a function of the pulse energy. Line indicates the fitting curve to Equation (3).

**Figure 9 polymers-12-00127-f009:**
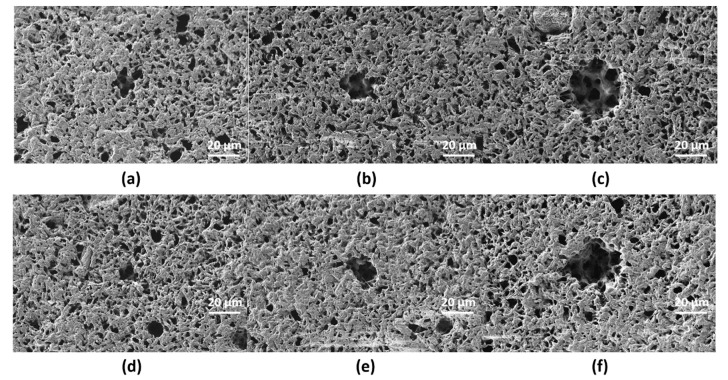
SEM images corresponding to single-shot craters at wavelengths of (**a–c**) 355 and (**d–f**) 532 nm on scl- and mcl-PHA blend (80/20) at different energies: *E* = 3.3 µJ (**a**), *E* = 7.4 µJ (**b**), *E* = 32.7 µJ (**c**), *E* = 4 µJ (**d**), *E* = 9.5 µJ (**e**), *E* = 61 µJ (**f**).

**Figure 10 polymers-12-00127-f010:**
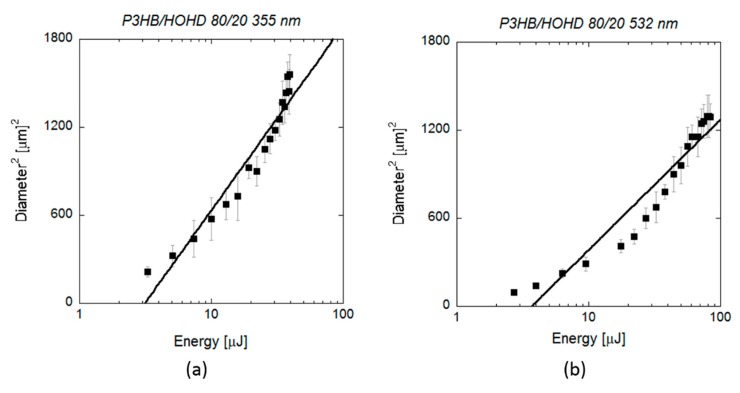
Squared diameter of the craters (black rectangles) produced by 10 ps pulses at a wavelength of (**a**) 355 and (**b**) 532 nm on scl-/mcl-PHA 80/20 blend and measured by SEM as function of the pulse energy. Lines indicate the fitting curves to Equation (2): (**a**) *ω*_0_ = (16.58 ± 0.08) µm and *E*_th_ = (3.1 ± 0.4) µJ, (**b**) *ω*_0_ = (13.96 ± 0.09) µm and *E*_th_ = (3.7 ± 0.6) µJ.

**Figure 11 polymers-12-00127-f011:**
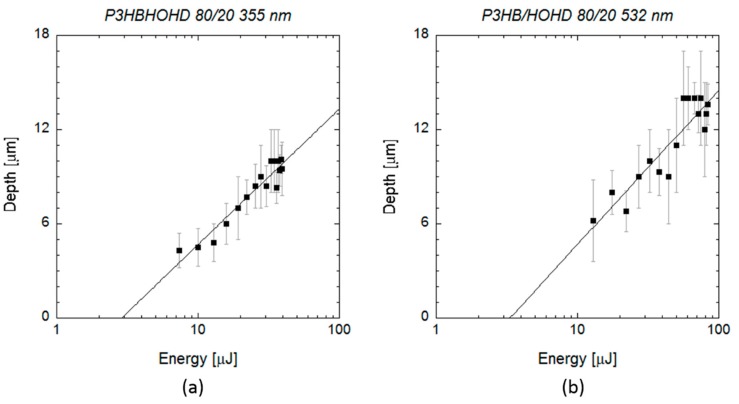
Data analysis of single pulse craters produced by 10 ps pulses at a wavelength of 355 and 532 nm on the blend. The graphs show the depth of the craters (black rectangles) measured by profilometry for wavelengths of (**a**) 355 nm and (**b**) 532 nm as a function of the pulse energy. Lines indicate the fitting curves to Equation (3): *E*_th_ = (2.9 ± 0.5) µJ, *α*_eff_ = (0.26 ± 0.02) µm^−1^ (UV), *E*_th_ = (3.3 ±1.1) µJ, *α*_eff_ = (0.23 ± 0.03) µm^−1^ (VIS)).

**Figure 12 polymers-12-00127-f012:**
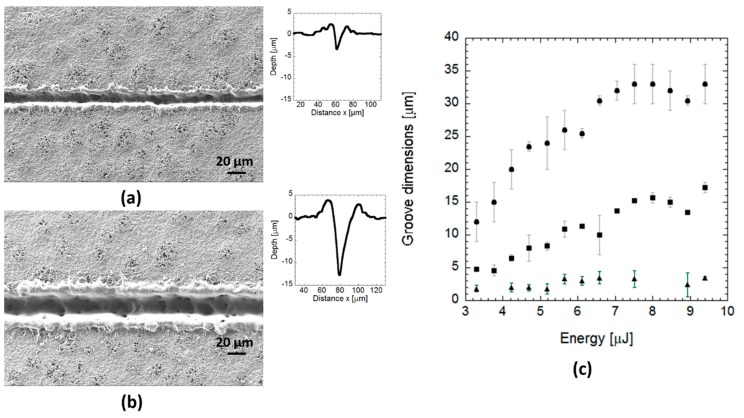
SEM images and topographic profiles corresponding to grooves machined by laser pulse overlapping on the blend at different pulse energy: (**a**) *E* = 3.8 µJ, (**b**) *E* = 9.4 µJ. (**c**) Evolution of groove width (circles), depth (squares), and height of the recast layers (triangles) with increasing pulse energy.

**Figure 13 polymers-12-00127-f013:**
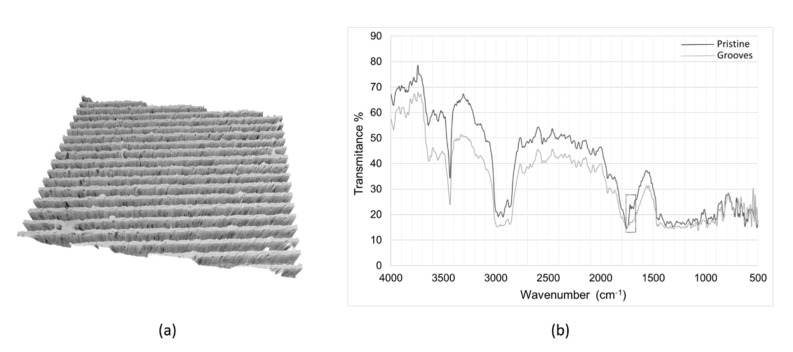
(**a**) 3D topography obtained by confocal microscopy on grooved blend samples of 40 × 40 mm, (**b**) infrared spectra of pristine (black line) and grooved (grey line) blend samples.

**Figure 14 polymers-12-00127-f014:**
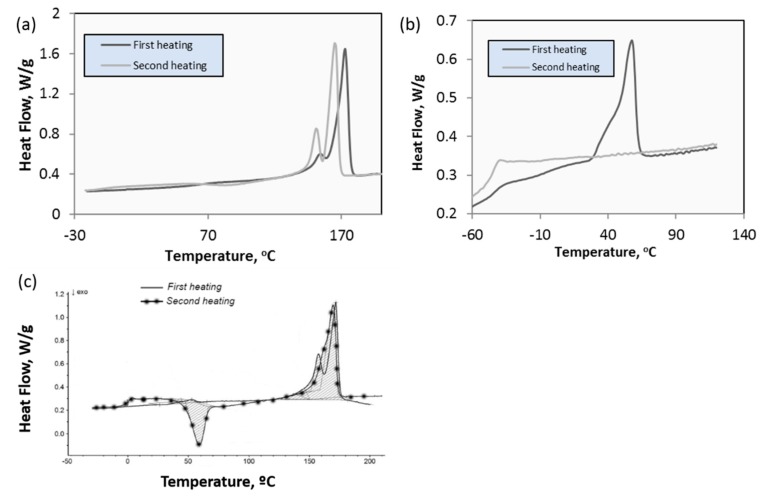
DSC curves of scl- (**a**) and mcl- (**b**) PHAs and the blend (**c**).

**Table 1 polymers-12-00127-t001:** Ablation thresholds (J/cm^2^) calculated from the threshold energies and beam waists obtained from fitting the crater diameters as a function of laser energy to the Equation (2). Ablation threshold in terms of laser fluency was calculated from the following well-known equation: $F_{peak}=\frac{2E_{p}}{\pi \omega_{0}^{2}}$.

Materials	Wavelength (nm)	Beam Waist (µm)	Threshold Energy (µJ)	Ablation Threshold (Jcm^−2^)
scl-PHA	355	16.43	4.00	0.9 ± 0.2
	532	13.60	6.80	2.3 ± 0.3
mcl-PHA	355	22.6	9.40	1.17 ± 0.08
	532	19.62	12.60	2.1 ± 0.2
Blend	355	16.58	3.10	0.72 ± 0.09
	532	13.96	3.70	1.2 ± 0.2

**Table 2 polymers-12-00127-t002:** Thermal properties of PHAs obtained by differential scanning calorimetry (DSC) (Figure 14): glass transition temperature (*T*_g_), crystallization temperature (*T*_C_), melting temperature (*T*_m_), enthalpy of fusion (Δ*H*_m_) and crystallization (Δ*H*_C_) and crystallinity degree (*X*_C_), which was calculated for P(3HB) using the formula *X*_c_ = $\frac{\Delta H_{m}}{\Delta H_{0}}\times 100$ and Δ*H*_0_ = 146 J/g [30].

	*T*_g_ (°C)	*T*_C_ (°C)	*T*_m_ (°C)	Δ*H*_m_ (Jg^−1^)	*X*_C_ (%)
P(3HB)	2.0 ± 0.1	60 ± 2	175	77 ± 8	≈ 52
P(3HO-3HD)	−43 ± 2	-	57	27 ± 2	-
P(3HB)/P(3HO-3HD)	−0.2 ± 2	59	170	63	-

**Table 3 polymers-12-00127-t003:** Material mechanical properties: molecular weights (*M*_n_ and *M*_w_), ultimate tensile strength (*σ***_U_**), Young’s modulus (*E*), and elongation at break (*ɛ***_b_**).

	*M*_n_, kDa	*M*_w_, kDa	*σ*_U_, MPa	*E*, GPa	*ɛ*_b_, %
P(3HB)	110 ± 5	800 ± 40	36 ± 4	1.4 ± 0.3	7 ± 2
P(3HO-3HD)	92 ± 5	380 ± 20	3.9 ± 0.7	1.1 ± 0.1	920 ± 100
P(3HB)/P(3HO-3HD)			22 ± 1	1.4 ± 0.2	27 ± 2

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
