# Peer review of "Picosecond Laser Ablation of Polyhydroxyalkanoates (PHAs): Comparative Study of Neat and Blended Material Response"

_polymers, 2020, doi:10.3390/polym12010127_

Round 1

Reviewer 1 Report

The manuscript describes surface modification of films from biodegradable natural polyhydroxyalkanoates (PHAs) produced by microorganisms. For this purpose the authors applied for the first time picosecond pulsed laser ablation. The modification of the topography of three types of membranes with different thermal and mechanical properties proceeded with non-significant effects on the chemical and microstructural properties.  These biomaterials are thus promising alternatives for scaffold manufacturing in the tissue engineering field or for food packaging. For this reason I recommend publication.      

Some minor corrections:

Page 9 Line 224 and 226: Please correct wavenumber units (cm-1). There are some differences in the IR spectra for example a peak near 550 cm-1. I would advice adding “Both grooved and pristine samples showed spectra with almost the same peaks” but this is unimportant.

Page 11 Line 280 no instead of not

I didn’t find Appendix 1

Reviewer 2 Report

This paper is another one of a series of papers by the same research group concerning micropatterning of biodegradable polyesters by laser ablation.  In this study, the authors studied the formation of crates and grooves on the surface of polyhydroxyalkanoates (PHAs) under different conditions, e.g., the wavelength and energy of laser light and the chemical structure of PHAs.  The experiments were carefully done, the results are sound, and the conclusion is supported by the data indicated.  Biodegradable polyesters are promising in a variety of fields, including tissue engineering and food packaging, and micropatterning is important for applications. Thus, the topic of this paper is of great importance.  In conclusion, I would like to recommend publication of this paper in Polymers after minor revision.  Specific points amended to are listed below.  

(1) Page 2, line 60.  "scl-PHA" and "mcl-PHA" should be defined at the first appearance.  

(2) The chemical structure of the PHAs employed should be indicated.  

(3) The basic characteristics (Mn and Mw/Mn) of the PHAs employed should be summarized in a table.  

(4) DSC curves of the PHAs should be indicated.  

(5) Appendix is missing.  

Reviewer 3 Report

A very interesting work about the effect of picosecond laser ablation of PHAs of different molecular weights and the blends between them. The subject of the paper is of very good interest, and the results presented highlight the potentiality of the proposed approach for surface modification of the PHAs.

Just some very minor issues to be addressed before publication:

In the abstract, row 18: “and the blend” should be added: “between short and medium chain length” Row 20: “ Picosecond pulsed laser modified the surface topography 19 without affecting the material properties” cannot be verified, since no comparison is shown between properties of PHA before and after ablation. This is part of the work which is actually missing, and I would suggest the authors to expand their work in order to verify how the ablation influences, for example, strain and strength at break. However, the paper is self-consistent as is. My suggestion can be sued for future works. Row 71: what do these acronyms mean? is this a polyhydroxy butyrate? Please also explain the meaning of P(3HO-3HD) Row 96: since this is deeply treated in a section, the meaning of pulse overlapping and the way this was achieved should be better explained In eq. 1) the left hand side is the diameter of the craters? If so, please be careful, since the diameter in row 121 is labeled as “D”. Table 3: data for the blend show no standard deviation
